# The L-Type Lectin-like Receptor Kinase Gene *TaLecRK-IV.1* Regulates the Plant Height in Wheat

**DOI:** 10.3390/ijms23158208

**Published:** 2022-07-26

**Authors:** Mamoudou Saidou, Zengyan Zhang

**Affiliations:** 1National Key Facility for Crop Gene Resources and Genetic Improvement, Institute of Crop Science, Chinese Academy of Agricultural Sciences, Beijing 100081, China; mamoudou_saidou@yahoo.fr; 2Department of Biological Sciences, Faculty of Science, University of Ngaoundere, Ngaoundere P.O. Box 454, Cameroon

**Keywords:** lectin receptor-like kinase, dwarf plants, virus-induced gene silencing, wheat (*Triticumaestivum*), sharp eyespot

## Abstract

Dwarfing is important for the production of wheat (*Triticumaestivum* L.). In model plants, receptor-like kinases have been implicated in signal transduction, immunity, and development. However, functional roles of lectin receptor-like kinases in wheat are poorly understood. In this study, we identified an L-type lectin receptor-like kinase gene in wheat, designated as *TaLecRK-IV.1*, and revealed its role in plant height. Real time quantitative PCR analyses indicated that *TaLecRK-IV.1* transcript level was lower in a dwarf wheat line harboring the *Rht-D1b* gene compared to its transcript level detected in a taller wheat line CI12633. Importantly, the virus-induced gene silencing results showed that silencing of *TaLecRK-IV.1* in the wheat line CI12633 led to dwarf plants. The results of the disease resistance test performed after the gene silencing experiment suggest no significant role of *TaLecRK-IV.1* in the resistance reaction of wheat line CI12633 to sharp eyespot. Gene expression analysis revealed that the transcript abundance of *TaLecRK-IV.1* was more up-regulated after the exogenous application of gibberellic acid and auxin, two development-related phytohormones, compared to the gene transcript levels detected in the control plants (mock treatment). These findings support the potential implication of *TaLecRK-IV.1* in the pathway controlling plant height rather than the disease resistance role, and suggest that *TaLecRK-IV.1* may be a positive regulator of plant height through the gibberellic acid and auxin-signaling pathways.

## 1. Introduction

Plants possess a communication system that allows them to receive and process information. The system does not only permit them to constantly detect the changes in environmental conditions that can lead to growth and developmental modulations but also helps them in detecting external menaces such as pathogen infections. The perception and transduction of signals from environmental cues to the plant intracellular milieu are mostly governed by the receptor-like kinases [1]. The genes encoding receptor-like kinases (RLKs) are found in great numbers and are largely distributed in plant genomes. For instance, over 600 RLK genes have been identified in *Arabidopsis thaliana* while rice (*Oryza sativa*) harbors more than 1000 genes [2]. In higher plants, RLKs form a vast protein family and lectin receptor kinases (LecRKs) are a group of RLK proteins characterized by an N-terminal extracellular lectin motif [3]. Based on the variability of the N-terminal lectin domains, this category has been sub-grouped into four classes, G-, C-, L-, and LysM-types [4]. Among the four classes, the L-type is the more abundant and is characterized by the presence of three distinct domains comprising a legume lectin-like extracellular domain, a transmembrane domain, and an intracellular kinase domain [5].

In *Arabidopsis*, LecRK-VI.2 protein is implicated in the pattern-triggered immune responses to bacterial pathogens [6]. The lectin receptor kinase genes *AtLecRK-IV.3* and *AtLecRK-I.9* were reported to induce the defense response to *Botrytis cinerea* and to *Phytophthora infestans*, respectively [3,7]. *LecRK-V*, an L-type lectin receptor kinase gene from a diploid wheat relative (*Haynaldia villosa*), confers broad-spectrum resistance to powdery mildew in transgenic wheat plants [8]. In addition to the implication of these genes against pathogen attacks, some reports mentioned their roles toward the environmental stresses and following the exogenous application of phytohormones (abscisic acid), while other studies have associated their roles with wounding [9] and senescence [10]. Although these genes have been enumerated in diverse biological processes, their exact roles remain to be clarified [3].

Wheat (*T. aestivum* L.) production is essential for global food security [11,12]. During the 1960s, the semi-dwarfing genes contributed to the green revolution [13]. Since then, the introduction of the dwarfing genes into wheat varieties has become one of the main objectives of wheat breeding programs worldwide [14]. To date, the most widely distributed dwarfing genes also known as ‘reduced height’ (*Rht*) in wheat are *Rht-B1b* and *Rht-D1b*. These alleles are the results of mutations that reduce the plant sensitivity to respond to gibberellins [15].

Gibberellins (GAs) are a class of tetracyclic diterpenoid phytohormones with a multitude of roles in plants [16]. GAs mediate different processes including stem elongation, seed germination, trichome development, leaf expansion, induction of flowering, and pollen maturation [17]. Mutant plants defective in GA signaling or bioactive GA production have altered growth and development-related processes including short plant stature [18]. GA biosynthesis goes through a long process and the later steps of the process are catalyzed by 2-oxoglutarate-dependent dioxygenase enzymes (*GA20ox* and *GA3ox*). These enzymes along with the *GA2ox* regulate the level of bioactive GA [18]. In contrast to the actions of these enzymes, there also exist the GA repressor proteins, known as DELLA proteins.

DELLA proteins act as transcription factors to regulate the GA signaling pathway. *SLR1* (SLENDER RICE1) is currently the well-known gene encoding a DELLA protein in rice (*Oryza sativa*) while five genes (*GA-INSENSITIVE* (GAI), *REPRESSOR OF GA1-3* (RGA), *RGA-LIKE1* (*RGL1*), *RGL2*, and *RGL3*) encoding proteins with similar functions were reported in the *Arabidopsis* genome [18].These proteins belong to the GRAS family. They possess a conserved domain at the C-terminal region while a conserved sequence of amino acid residues Asp–Glu–Leu–Leu–Ala that plays a regulatory role is found at their N-terminal region. Currently, it is established that DELLAs restrain plant growth and this is in opposition to the action of GA, which promotes growth [18].

In cereal crops such as wheat and rice, the dwarfing genes decrease the plant response to GA-mediated growth. Thus, they repress the stem elongation, resulting in dwarfism. Hence, plants harboring the reduced height (*Rht*) genes are dwarf but this stature improves the plant resistance to lodging [19]. The *Rht* genes significantly contributed to wheat production in the 1970s [20]. Hence, they are of great interest in wheat breeding to improve yield. Although these genes are extremely useful to address some issues regarding wheat production, their transfer into elite cultivars remains crossing-based, necessitating considerable fieldwork. Thus, alternative methods of developing dwarf wheat varieties are still needed. The development of the dwarf cultivars for yield improvement can also contribute to reducing the production loss caused by major wheat diseases, such as sharp eyespot. The sharp eyespot disease is caused by *Rhizoctonia cerealis* and impacts wheat yield in many regions of the world. For instance, in China, it is estimated that more than 6.66 million hectares of wheat were affected by sharp eyespot since 2005 [21,22,23,24].

Considering the number of genes encoding lectin-like receptor protein kinases in species such as *Arabidopsis* and rice, these genes should have rich biological functions for crop improvement as they are potentially believed to play roles in growth regulation and stress responses [3]. In comparison to *Arabidopsis* and rice, few works have been devoted to wheat lectin-like receptor kinase genes. To date, most of the literature on the functions of lectin-like receptor kinase genes concerns the transduction of extracellular oligosaccharide signals [3]. Studies documenting the implication of lectin-receptor kinase genes on plant development are limited.

In this study, we cloned a gene in wheat encoding a LecRK-IV.1, named *TaLecRK-IV.1*, and found that silencing of *TaLecRK-IV.1* led to dwarf plants in a wheat line CI12633. Test for disease resistance following *TaLecRK-IV.1*-silencing revealed no impairment of resistance of wheat line CI12633 to sharp eyespot despite the reduction of *TaLecRK-IV.1* transcripts. The results suggest that *TaLecRK-IV.1* is involved in the regulation of plant growth but not in disease resistance. *TaLecRK-IV.1* was induced by the exogenous application of gibberellic acid (GA) or auxin. This gene may be useful in wheat breeding to develop alternative dwarf varieties without the need of employing the previous dwarfing alleles.

## 2. Results

### 2.1. Cloning of TaLecRK-IV.1 Gene Encoding an L-Type Lectin Domain Containing Receptor Kinase IV.1

The annotated sequence of TRIAE_CS42_4AL_TGACv1_288345_AA0945480 (https://plants.ensembl.org/Triticum_aestivum/ (accessed on 16 February 2018)) corresponding to *TaLecRK-IV.1* was identified by BLAST search using a short putative gene sequence obtained following an RNA-seq experiment. The primer pair [QF: 5′-ATACTACTGAGTTTTCTTTTGTCC-3′ and QR: 5′-TAGGGCTGTGATTCTGAGGC-3′, Appendix A] were then designed and used to amplify the predicted full length of the gene based on the database information. The amplicon from the cDNA of the wheat cultivar (cv.) CI12633 was cloned, sequenced, and analyzed; and the sequence analysis indicated the characteristic of a protein-coding gene containing an open reading frame (ORF). The deduced protein from the cloned sequence consisted of 676 amino acid residues. The amino acids sequence of *TaLecRK-IV.1* protein contains a Lectin-like superfamily domain at the N-terminal region, an ATP binding site, and a protein kinase superfamily domain at the C-terminal region. Based on the classification of lectin domains, the protein belongs to the L-type (lectin-like) group, the most abundant group of lectin receptor kinases [4]. By using the amino acid sequence of *TaLecRK-IV.1* as query, BLAST search in the protein database found many L-type domains containing receptor kinase IV.1 proteins from different species, such as wheat (*T. aestivum*), barley (*Hordeum vulgare*), rice (*Oryza sativa*), maize (*Zea mays*), brachypodium (*Brachypodium distachyon*), foxtail millet (*Setaria italica*), and sorghum (*Sorghum bicolor*). The predicted TaLecRK-IV.1protein sequence shared 61–93% identities with those of wheat, barley, rice, maize, brachypodium, foxtail millet, and sorghum (Appendix A). In Arabidopsis, the L-type LecRK proteins were classified into nine clusters and nine clades on the basis of phylogenetic relationships [4]. BLAST and phylogenetic tree analyses showed that TaLecRK-IV.1 fell into the cluster IV, clade 1, and TaLecRK-IV.1 was closer to *Aegilops tauschii* and *Triticum aestivum* L-type LecRks (Figure 1).

### 2.2. Silencing of TaLecRK-IV.1 Leads to Dwarf Plants

CI12633, a tall wheat cultivar [25], a was employed in the VIGS experiment. Seeds were planted in pots, and seedlings at two- to three-leaf stage were inoculated with a mixture of BSMV RNA (α, β, and γ:00 or γ:*TaLecRK-IV.1*). Two types of mixture derived from the in-vitro transcription were set up. One contained the BSMV RNA α, β, and γ:00 and was used to inoculate CI12633 seedlings considered as control. Another mixture containing the BSMV RNA α, β, and γ:*TaLecRK-IV.1* was employed to infect other CI12633 seedlings for the gene silencing purpose. About 10 days post-inoculation, symptoms of BSMV infection were observed on the newly developed leaves of more than 90% of the inoculated plants and were seen on both groups (control and *TaLecRK-IV.1*-silenced plants). Based on the specific PCR fragment for BSMV coat protein gene, the presence of BSMV was detected in all plants exhibiting the phenotype characteristic of BSMV infection. Importantly, the RT-qPCR analyses showed that compared to control, *TaLecRK-IV.1* transcriptional level was considerably reduced in BSMV:*TaLecRK-IV.1*-infected wheat plants (Figure 2). *TaLecRK-IV.1* nucleotide sequence shared 92% similarity with another gene located on wheat chromosome 7AS, namely 7AS-homolog, and both were supposed to be homologous genes. The 224 bp fragment used to construct the BSMV vector to silence *TaLecRK-IV.1* has 88% similarity with the corresponding fragment of the 7AS-homolog. Thus, the transcript of the 7AS-homolog was also analyzed by RT-qPCR and the result indicated that this gene was also down-regulated (Figure 2) but to a lesser extent as compared to *TaLecRK-IV.1*. In accordance with this result, a previous study reported that endogenous genes sharing high homology with the fragment beard by the BSMV can be silenced by VIGS, even if the inserts in the BSMV vectors are from other species [26]. 

Following the observation of successful BSMV infection and the gene expression levels of *TaLecRK-IV.1* and its 7AS-homolog in wheat CI12633, the monitoring of the plants has permitted to realize that the plants infected with BSMV:*TaLecRK-IV.1* displayed a dwarf phenotype as compared to those inoculated with BSMV:00 (control, Figure 3). In the previous experiments in our laboratory [27,28,29,30,31], dwarf phenotype was not observed when the wheat cultivar CI12633 was subjected to the VIGS experiments. Moreover, CI12633 is a tall cultivar [25], which harbors none of the well-known and largely distributed dwarfing genes (*Rht-B1b* and *Rht-D1b*). Taken togather, the results suggested that silencing of *TaLecRK-IV.1* led to reduced plant height, and that *TaLecRK-IV.1* might positively regulate plant height.

### 2.3. The Expression of TaLecRK-IV.1 in the Wheat Line with Rht-D1b Gene

In order to hypothesize that the reduced expression of *TaLecRK-IV.1* and/or its 7A-homolog gene affects plant height in wheat, we analyzed by RT-qPCR the transcript abundance of these genes in Xinong223, a wheat line harboring the *Rht-D1b* dwarfing gene. It was found that the expression level of *TaLecRK-IV.1* was lower, while that of the 7A-homolog gene was higher in the wheat line carrying the *Rht-D1b*. The transcript level of *TaLecRK-IV.1* was significantly lower in the dwarf wheat line compared to its expression detected in wheat cultivar CI12633, whereas an opposite expression difference (high expression) was found for the 7A-homolog gene in the *Rht-D1b* carrying wheat line as compared to that of the control (CI12633) (Figure 4). The results suggested that in the presence of *Rht-D1b*, the transcript of *TaLecRK-IV.1* was not abundant, probably due to the action of the *Rht* allele. While the expression level of *TaLecRK-IV.1* in the *Rht-D1b*-carrying wheat line was low, that of its 7A homologue gene was elevated in the same wheat genotype (Xinong223) as compared to the transcript abundance detected in the control (CI12633). The low expression of *TaLecRK-IV.1* in a wheat line with *Rht-D1b* (dwarf line) was in agreement with the hypothesis that the product of this gene may be implicated in the pathway affecting plant height. These data suggested that the expression of *TaLecRK-IV.1* might be influenced by the presence of the *Rht-D1b*, which may act upstream of *TaLecRK-IV.1*. Also, the data imply that when the expression of *TaLecRK-IV.1* is not altered, it contributes to normal plant height, whereas its silencing leads to dwarf plants.

### 2.4. Relative Expression of GA Biosynthetic Enzymes TaGA20ox and TaGA3ox in Rht-D1b-Carring Wheat Line and TaLecRK-IV.1-Silenced CI12633 Plants

To explore if the silencing of *TaLecRK-IV.1* affects the expression of these enzymes, and led to dwarf phenotype, we analyzed the relative abundances of *TaGA20ox* and *TaGA3ox* genes in *TaLecRK-IV.1*-silenced CI12633 plants. The RT-qPCR results showed that as compared to control plants (CI12633 infected with BSMV:00), *TaGA20ox* expression was moderately elevated in the *TaLecRK-IV.1*-silenced plants. Likewise, *TaGA20ox* was highly expressed in Xinong223 (*Rht-D1b*) wheat line. On the other hand, the relative expression of *TaGA3ox* did not show any significant difference in both *TaLecRK-IV.1*-silenced plants and Xinong223 (*Rht-D1b*) wheat line as compared to the BSMV:00-infected CI12633 (control) plants (Figure 5). The relative high level of *TaGA20ox* in the silenced plants might be related to the silencing of *TaLecRK-IV.1*.

### 2.5. TaLecRK-IV.1 Is Expressed in Response to Exogenous GA and Auxin Treatments

We further investigated if *TaLecRK-IV.1* responds to exogenous hormone treatments. Upon GA treatment, *TaLecRK-IV.1* transcript level was considerably increased from 1 to 3 h then declined at 6 h, and finally the induction resumed at 12 h before reaching the highest level at 24 h (Figure 6A). Following exogenous auxin application, *TaLecRK-IV.1* expression was first down-regulated about 30 min post-treatment. After that, the expression regained approximately to a normal level about 1 h before being up-regulated between 3 to 6 h post-treatment with auxin. After 6 h, the gene transcript declined considerably between 12 and 24 h post-treatment (Figure 6B).

Towards the exogenous ET and SA stimuli, the expression trend of *TaLecRK-IV.1* was variable. Overall, ET and SA considerably up-regulated the expression of *TaLecRK-IV.1* about 1 h post-treatment. Afterward, ethylene-induced *TaLecRK-IV.1* expression remarkably declined about 3 h post-application and the expression trend was continuously decreasing till 24 h post ethylene treatment (Figure 7A). From the hormone treatments, we noted that auxin-induced expression started to differentially increase when the up-regulation caused by ethylene started decreasing about 3 h post-treatment. Considering the expression trend following salicylic acid treatment, the gene was down-regulated throughout the time points except at 1 h where a somewhat increase in the expression of *TaLecRK-IV.1* was observed as compared to that of control (Figure 7B). This expression trend may be explained by the fact that SA mostly induces the expression of the defense-related genes.

### 2.6. Resistance Responses to Sharp Eyespot of the Plants Subjected to the VIGS Experiment

As *TaLecRK-IV.1* was among the differentially expressed genes (DEG) based on the RNA-seq results, the gene was predicted to contribute to the disease resistance reaction. To elucidate whether its up-regulation was an active plant defense response, *TaLecRK-IV.1* was first silenced by VIGS in wheat line CI12633 and then the plants were inoculated with *Rhizoctonia cerealis* two weeks after *BSMV* infection. The results of the disease resistance test have revealed no significant difference between the level of resistance of the control plants (infected with *BSMV*:00) and those inoculated with the *BSMV*:*TaLecRK-IV.1* vector construct (Table 1). This result suggested that *TaLecRK-IV.1* may not be the major gene required in the resistance pathway to counter sharp eyespot infection. Thus, its up-regulation as revealed by RNA-seq could be related to the perturbation of the steady-state of the plants initially challenged by sharp eyespot.

### 2.7. TaLecRK-IV.1 Is Assigned to Chromosome 4A

To confirm the location of *TaLecRK-IV.1* on wheat chromosome 4A, a specific pair of primers to the gene was designed and tested on a set of DNA of Chinese Spring nulli-tetrasomic (N-T) lines. The PCR assay detected a single band product of expected size (~630 bp) in Chinese Spring but failed to amplified the target fragment of *TaLecRK-IV.1* in nulli 4A-tetra 4B and in nulli 4A-tetra 4D (N4A-T4B & N4A-T4D, Figure 8). This result indicated that *TaLecRK-IV.1* was located on wheat chromosome 4A.

## 3. Discussion

In this study, *TaLecRK-IV.1* was cloned and its expression profile was investigated. The functional role of *TaLecRK-IV.1* was analyzed by the mean of VIGS, and the results showed that silencing of *TaLecRK-IV.1* led to dwarf plants in CI12633. The disease infection test revealed no significant difference between the control plants and those subjected to the gene-silencing experiment. This means that *TaLecRK-IV.1* may be involved in the pathway regulating plant height rather than the one mounting the defense reaction against *Rhizoctonia cerealis* infection.

CI12633 is a tall wheat cultivar and does not carry any known dwarfing gene. Interestingly, *TaLecRK-IV.1* transcript level was found to be relatively low in the dwarf wheat line harboring the *Rht-D1b* as compared to its level in CI12633. Taken together, the low transcript level of *TaLecRK-IV.1* in the *Rht-D1b*-harboring dwarf wheat line and the reduced height caused by its silencing support the hypothesis that this gene could positively regulate plant height and might act downstream of the *Rht-D1b*. Thus, it is hypothesized that in the presence of a dwarfing gene, *TaLecRK-IV.1* expression might be repressed, which leads to short plant stature conferred by the action of the *Rht* allele, whereas in the absence of an *Rht* gene, *TaLecRK-IV.1* may contribute to normal plant height leading to tall plants. In *Arabidopsis*, lectin-like receptor kinases are believed to potentially participate in the regulation of plant growth and stress response [3]. Thus, *TaLecRK-IV.1* appears to be the first reported LecRK in plant potentially involved in mediating the processes controlling plant height.

In plants, development is mainly under the control of the growth-mediated phytohormones of which the roles of auxin and gibberellins are well characterized. Here, RT-qPCR analysis of *TaLecRK-IV.1* transcription revealed that the gene was induced upon GA and auxin treatments. Following the GA treatment, the gene was slightly down-regulated just after treatment (30 min post-application) before the transcripts became more abundant about 1 h post-treatment and afterward, except at 6 h post GA application where the expression declined to a comparable level to that of control. Under GA treatment, the highest level of the gene transcripts was detected at 24 h post-treatment. Similar to GA-induced up-regulation of *TaLecRK-IV.1*, auxin treatment gradually up-regulated the transcript of *TaLecRK-IV.1* from 1 h to 6 h post auxin application. The other time points were marked by a significant decrease in the expression of *TaLecRK-IV.1*,notably at 0.5 h and 24 h. The expression started decreasing after 6 h and fell to the lower level about 24 h post-treatment. These results implied that *TaLecRK-IV.1* may be a positive regulator of plant height, and its action might be through the GA and auxin signaling pathways.

Silencing of *TaLecRK-IV.1* produces dwarf plants comparable to the phenotypic effects of the *Rht* genes. Thus, it was hypothesized that *TaLecRK-IV.1* may be either a GA signaling gene or it acts on the GA biosynthesis pathway. To establish a relation between the role of *TaLecRK-IV.1* and the GA biosynthesis, the transcript levels of *TaGA20ox* and *TaGA3ox*, two enzymes that catalyze the later steps in the GA biosynthetic pathway, were analyzed by RT-qPCR. The RT-qPCR results showed that the relative expression of *TaGA20ox* was higher in *TaLecRK-IV.1*-silenced plant than the transcript levels detected in control (CI12633), whereas the transcripts of *TaGA30x* was not affected by the effects of *TaLecRK-IV.1*-silencing. These results suggest that silencing of *TaLecRK-IV.1* may cause the accumulation of *TaGA20ox* transcripts but without affecting the expression of *TaGA3ox*. This implies that *TaLecRK-IV.1* may be involved in the regulation of plant height through a mechanism affecting the expression of *TaGA20ox*.

Increased expression of the GA oxidase genes leading to dwarfism was reported in some previous studies [32,33,34,35,36]. Ford et al. [32] reported that *Rht18* semidwarfism in wheat was caused by the increased expression of the *GA2oxA9* (GA 2-oxidaseA9) that led to the reduction of the bioactive GA1 content through the conversion of GA12 or GA53 into the inactive forms GA110 or GA97, respectively. Appleford et al. [34] documented that the ectopic expression of *PcGA2ox1* from bean (*Phaseolus coccineus*) in wheat resulted in low GA content and led to strong plant height reduction but with other effects on the growth habit. Similarly, in rice, severe height reduction was observed following the overexpression of the C19 *OsGA2ox1* gene [35,36], while a similar experiment overexpressing the C20 *OsGA2ox6* or *OsGA2ox9* yielded plants with moderate height reductions without effects on the date of anthesis or grain yields [37,38]. These studies are in accordance with the hypothesis that the accumulation of the GA oxidase enzyme is associated with a decrease in plant height. Nevertheless, the GA-responsive *Rht18* is a dwarfing gene that affects the biosynthesis of GA1 through the up-regulation of *TaGA2oxA9*, whereas none of the gene family of the lectin receptor-like kinase was reported to be implicated in plant height control. We presume that the increased transcription of the *TaGA20x* detected in the *TaLecRK-IV.1*-silenced plants may lead to a comparable action to that of *Rht18*, thus preventing the accumulation of bioactive GA, and as result, leading to a height reduction. Further investigation will clarify the mechanism through which *TaLecRK-IV.1* acts in the pathway controlling plant height.

## 4. Materials and Methods

### 4.1. Plant Materials, Growth Conditions and Treatments

Two wheat (*T. aestivum*) genotypes, including CI12633 and Xinong223 (a dwarf wheat variety harboring the *Rht-D1b* formerly known as *Rht2*), were used in this experiment. The dwarf wheat line was obtained from the laboratory of Professor Yin-Gang Hu (Northwest A&F University, Xianyang, China). The plant materials were grown under a 14 h light (~22 °C)/10 h dark (~10 °C) cycle in greenhouse.

Seedlings of the wheat cultivar CI12633 at three-leaf stage were sprayed with exogenous chemical precursors of gibberellic acid, auxin, ethylene, and salicylic acid. Leaf tissues corresponding to each treatment were separately collected at the following time points: 0, 0.5, 1, 3, 6, 12, and 24 h post chemicals application.

### 4.2. RNA Isolation and cDNA Synthesis

RNA was extracted from the collected leaf tissues using a commercial RNA isolation kit, TriZol (Invitrogen, Waltham, MA, USA). Three biological replicates were sampled from the control plants, plants used in the gene-silencing experiment, as well as from those employed in phytohormone treatment analyses. The integrity of the RNA was verified by agarose gel. Reverse transcription (RT) of RNA into cDNA was performed following a two-steps protocol using a Superscript II RT kit (TaKaRa, Kusatsu, Japan).

### 4.3. Cloning and Sequence Analysis of TaLecRK-IV.1

To clone the full length of the gene *TaLecRK-IV.1*, a pair of primers was designed based on the putative sequence of TRIAE_CS42_4AL_TGACv1_288345_AA0945480, a predicted protein coding gene from the new reference sequence of the Chinese Spring annotated by the Earlham institute (formerly TGAC, https://plants.ensembl.org/Triticum_aestivum/ (accessed of 16 February 2018)). The PCR product amplified using the following primers ATACTACTGAGTTTTCTTTTGTCC-F and TAGGGCTGTGATTCTGAGGC-R corresponding to the expected size was cloned and then sequenced. The obtained sequence was analyzed and its open reading frame (ORF) spanning more than 2 kb was deduced.

### 4.4. Virus-Induced Gene Silencing of TaLecRK-IV.1

Barley Strip Mosaic Virus (BSMV) subgenome RNAs described by Holzberg et al. [26] were employed in this study to construct vectors for the silencing experiment. In the BSMV RNA, the subgenome γ was first double digested with Nhe I restriction enzyme and then used for sub-cloning with a 224 bpcDNA fragment of *TaLecRK-IV.1*, which was inserted in antisense orientation into the subgenome γ. The resulting recombinant γ construct was confirmed by sequencing. Subsequently, plasmids from the recombinant γ vector as well as from the BSMV α and BSMV β were isolated and linearized by restriction enzyme digestion (Mlu I for α and γ, and Spe I for β). The linearized plasmid DNAs were transcribed in vitro to RNAs following the instruction manual of RiboMAX Large Scale RNA Production-T7 kit (Promega). An equal volume (1:1:1) of RNAs derived from the transcription of linearized α, β, and γ was first mixed and then the FES buffer was added; and the mixture was used for transfecting seedlings of the wheat line CI12633 at the three-leaf stage as described by Panwar et al. [39]. After inoculation, these seedlings were immediately sprayed with RNA-free water to improve the humidity and then taken into a controlled environment where they were misted regularly for 48 h at 20~22 °C. Finally, the seedlings were kept under high humidity for another 24 h before transferring to a glass house.

### 4.5. Real Time PCR Analysis

RT-qPCR was performed in a 96-well plate ABI 7500 real-time PCR equipment of Applied Bio-systems. The total reaction volume was 20 µL and included 10 µL SYBR green (Premix Ex Taq II), 0.4 µL ROX Reference Dye II from TaKaRa, 0.5 µL of each primer, 4 µL of template cDNA, and 4.6 µL ddH_2_O. The reaction for each gene was replicated three times. The PCR amplification was performed according to the program defined by TaKaRa in the manual for the SYBR product. Specific primers for the genes of interest (Appendix A) were designed based on the information of the reference sequences of the target genes and after BLAST search analyses. The specificities and efficiencies of the primers were evaluated in a preliminary assay prior to the gene expression analysis. The gene expression was calculated using the relative quantification method employing the 2^−∆∆CT^ formula [40]. *TaActin* was used as an endogenous control. The quantitative PCR was employed to analyze the expression of *TaLecRK-IV.1* in the plants employed in the silencing experiment and in those subjected to phytohormone treatments.

### 4.6. Test for Susceptibility to Sharp Eyespot

Two weeks post-BSMV infection, sterilized wheat kernels cultured to propagate a sharp eyespot and on which *Rhizoctonia*
*cerealis* was well-developed were used to inoculate CI12633 wheat plants.

The fungal pathogen *Rhizoctonia cerealis* isolate, R0301, the most frequent and virulent in Jiansu province (China), was obtained from Jiansu Academy of Agricultural Sciences. The pathogen isolate was cultured on potato dextrose agar and then propagated on sterilized wheat kernels and toothpicks at room temperature for inoculation purposes. The plants of the wheat genotype CI12633 were inoculated at the bottom sheath. After inoculation, the relative humidity was increased to facilitate the infection of the inoculated plants. The test aimed to evaluate the resistance reaction of the plants employed in the gene-silencing experiment (i.e plants infected with BSMV:*TaLecRK-IV.1* carrying a target gene insert) as compared to the control plants inoculated with BSMV:00 (which has no foreign gene fragment). Through this test of susceptibility, the involvement of the gene targeted for silencing in the resistance pathway can be inferred if its down-regulation leads to more vulnerability of the host plant following pathogen challenge. The phenotyping evaluation of the disease infection was done at 21 days post-inoculation.

## 5. Conclusions

This study suggests that the L-type lectin-like receptor kinase gene *TaLecRK-IV.1* is a positive regulator of plant height and its silencing led to dwarf plants in CI12633. This gene may act downstream of the *Rht* alleles and play roles through the pathway governing the synthesis of bioactive GA. *TaLecRK-IV.1* will be useful in wheat breeding programs as it can be targeted by the gene-editing tools for developing alternative dwarf varieties without the necessity of employing the *Rht* genes.

## Figures and Tables

**Figure 1 ijms-23-08208-f001:**
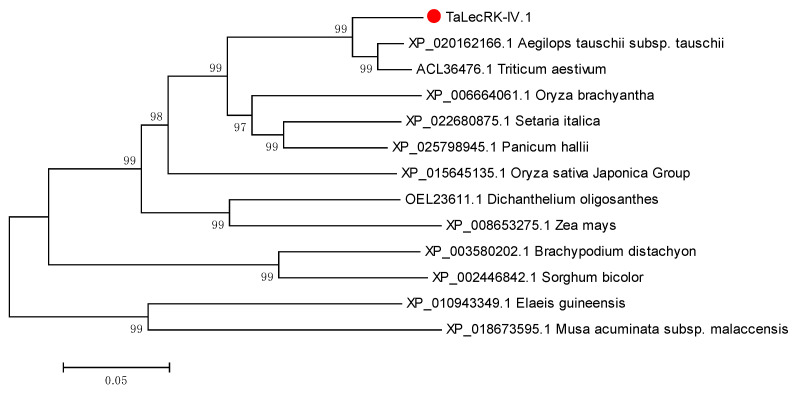
Phylogenetic analysis of the predicted plant proteins in the Genbank and *TaLecRK-IV.1* protein deduced from the coding sequence of *TaLecRK-IV.1*. Red dot indicates *TaLecRK-IV.1* protein. Database accession numbers and the name of the species are shown. The phylogenetic tree was constructed using the neighbor-joining phylogeny of MEGA 5.0.

**Figure 2 ijms-23-08208-f002:**
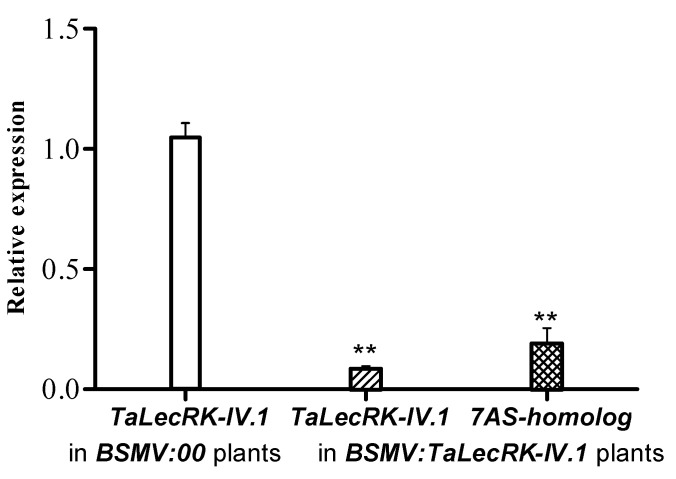
Analysis of *TaLecRK-IV.1* expression in CI12633 after the gene-silencing experiment. BSMV:00 plants were infected with the empty BSMV vector and used as control, whereas BSMV:*TaLecRK-IV.1* plants were inoculated with BSMV vector carrying a fragment of *TaLecRK-IV.1* for gene silencing. The relative gene expression (comparative 2^−∆∆C^) method was employed. The endogenous reference gene *TaActin* was used to normalize the expression level of *TaLecRK-IV.1*. Double asterisks indicate significant expression difference at 0.01 probability level as compared to the relative expression in control by using the Student’s *t*-test.

**Figure 3 ijms-23-08208-f003:**
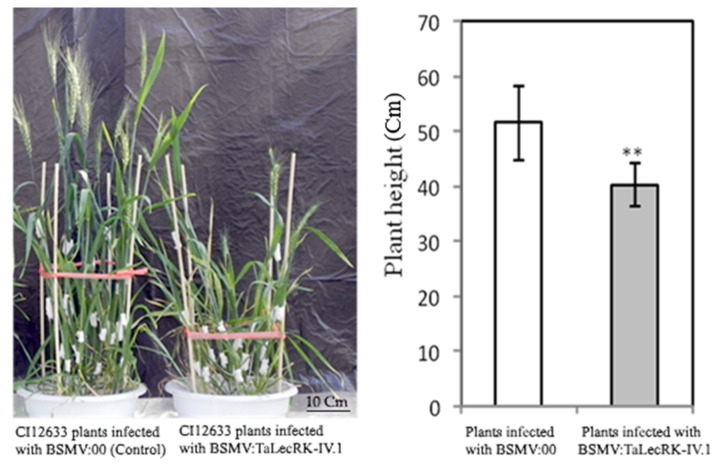
Phenotypes (**left**) and average heights (**right**) of the wheat CI12633 plants subjected to *TaLecRK-IV.1*-silencing experiment. CI12633 plants were inoculated either with empty BSMV vector (γ:00) and used as control or infected with BSMV vector carrying a fragment of *TaLecRK-IV.1* (γ:TaLecRK-IV.1) for the gene-silencing purpose. The test of comparison for the average plant height results from the Student’s *t*-test (double asterisks indicate significant difference at 0.01 probability level).

**Figure 4 ijms-23-08208-f004:**
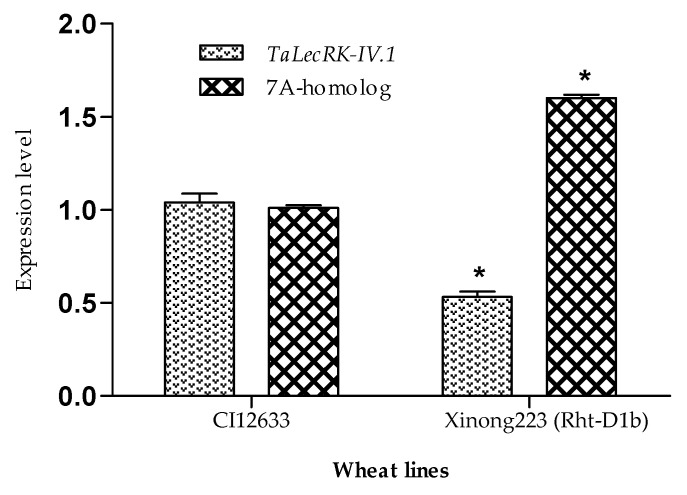
RT-qPCR analysis of the expression of *TaLecRK-IV.1* and its 7A-homolog gene in wheat cv. CI12633 and in wheat lineXinong223 (*Rht-D1b*). For comparison, the expression of *TaLecRK-IV.1* and its paralogue in CI12633 was employed as control. The relative gene expression (comparative 2^−∆∆C^) method was employed. The endogenous reference gene *TaActin* was used to normalize the expression levels of *TaLecRK-IV.1* and its 7A-homolog. Asterisks indicate significant expression difference at 0.05 probability level as compared to the relative expression in control by using the Student’s *t*-test.

**Figure 5 ijms-23-08208-f005:**
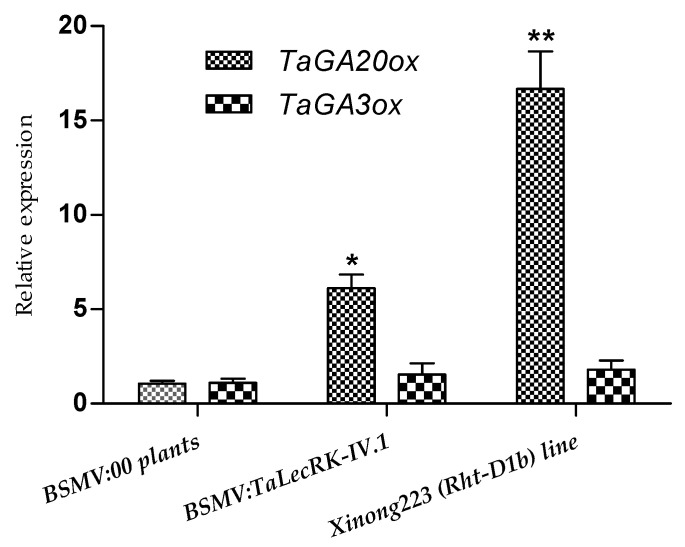
Expression of the dioxygenase genes (*TaGA20ox* and *TaGA3ox2*) in CI12633 plants infected with BSMV:00 (control), BSMV:*TaLecRK-IV.1* (*TaLecRK-IV.1*-silenced plants), and in Xinong223 (*Rht-D1b*) plants. The relative gene expression (comparative 2^−∆∆C^) method was used and *TaActin* was considered as the reference gene to normalize the expression level of *TaGA20ox* and *TaGA3ox2*. Asterisks indicate significant expression difference at 0.05 (*) or 0.01 (**) probability level as compared to the relative expression in control based on the Student’s *t*-test results.

**Figure 6 ijms-23-08208-f006:**
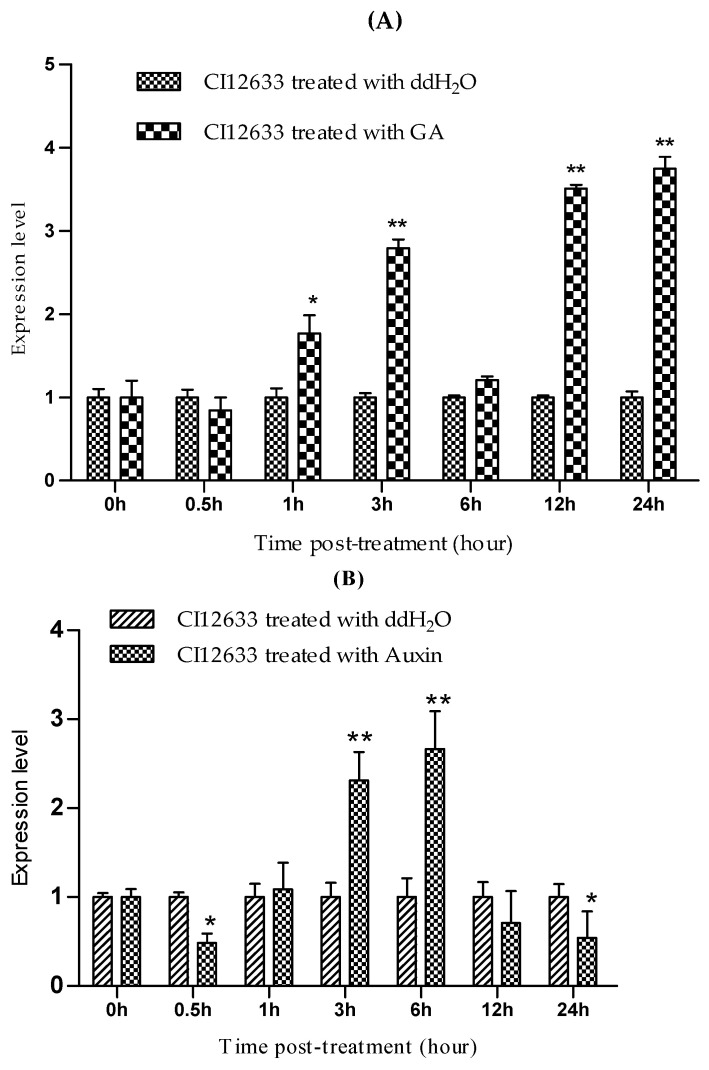
Expression trends of *TaLecRK-IV.1* in CI12633 wheat plants treated with exogenous GA (**A**) and Auxin (**B**). Plants treated with sterile ddH_2_O were considered as control. The relative gene expression was quantified using the comparative 2^−∆∆CT^ method, and *TaActin* was used as the reference gene to normalize the expression level of *TaLecRK-IV.1*. The Student’s *t*-test was used to compare the expression data and asterisks indicate significant expression difference at 0.05 (*) or 0.01 (**) probability level.

**Figure 7 ijms-23-08208-f007:**
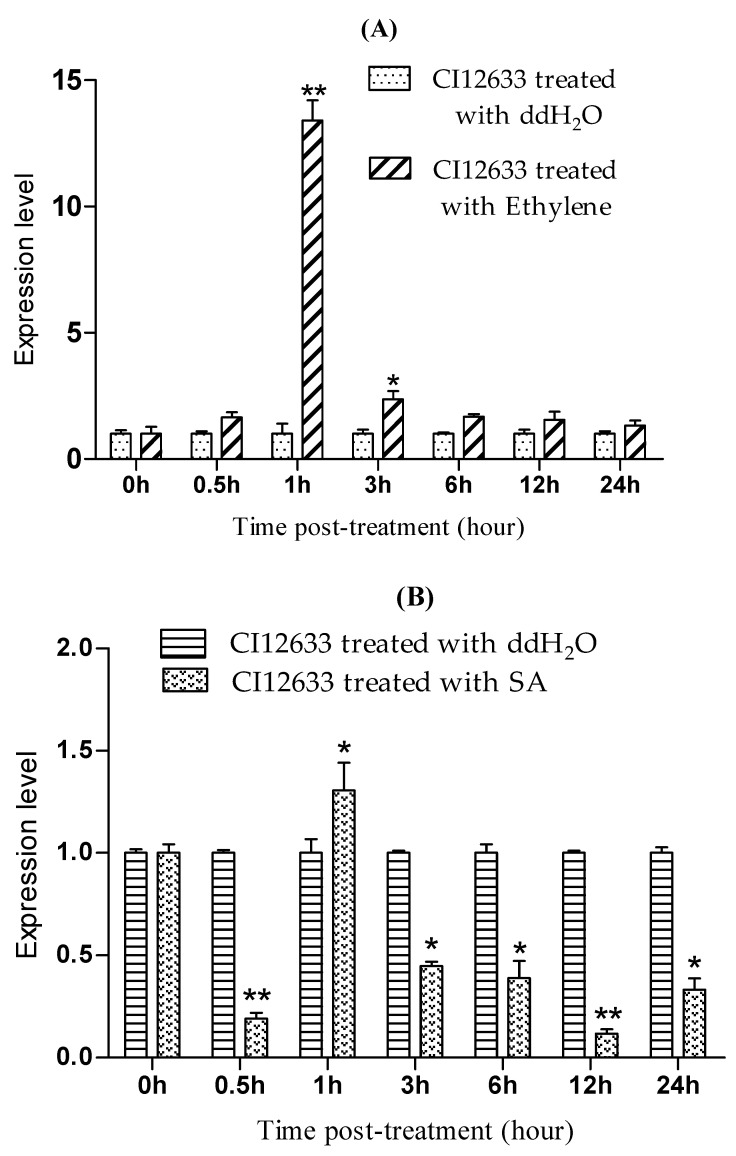
Expression trends of *TaLecRK-IV.1* in CI12633 wheat plants treated with exogenous ET (**A**) and SA (**B**). Plants treated with sterile ddH_2_O were considered as control. The relative gene expression was quantified using the comparative 2^−∆∆CT^ method and *TaActin* was used as the reference gene to normalize the expression level of *TaLecRK-IV.1*. Asterisks indicate significant difference (* *p* < 0.05 or ** *p* < 0.01) based on the Student’s *t*-test.

**Figure 8 ijms-23-08208-f008:**
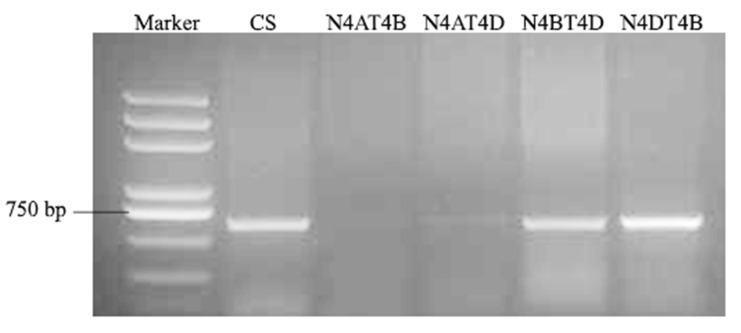
PCR assay to confirm the chromosomal location of *TaLecRK-IV.1* based on the amplification results of the gene-specific primer pair tested on a set of Chinese Spring (CS) nulli-tetrasomic lines.

**Table 1 ijms-23-08208-t001:** Average disease index of control (BSMV:00 infected) and silenced (BSMV:*TaLecRK-IV.1* infected) CI12633 plants and test of comparison of the two means by Student’s *t*-test.

Categories	Mean Disease Index	Student’s *t*-Test
Control plants	27.43	0.848
*TaLecRK-IV.1*-silenced plants	25.00	

## Data Availability

Not applicable.

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
