# Peer review of "The L-Type Lectin-like Receptor Kinase Gene TaLecRK-IV.1 Regulates the Plant Height in Wheat"

_ijms, 2022, doi:10.3390/ijms23158208_

Round 1
Reviewer 1 Report
In this work, the authors identified an L-type lectin receptor-like kinase gene TaLecRK-IV.1 in wheat and studied its functions in regulating wheat height. Overall, this study is well designed and clearly presented. The biological conclusions can be supported by the experiments including gene silencing, time-course expression analysis, etc. This work provide a clue to improve the plant height in wheat in future studies, and will benefit the cope improvement in the future. The manuscript will further be improved if the some small concerns below can be addressed before publication.
1. In Figure 3, left panel, the scale bar should be given.
2. In Figure 5 and 6, the statistical test method (t-test or others?) should be given for the significance.
3. In Figure 8, the band size label of the Marker should be given.
Author Response
- Minor spell check required.
Answer: Minor mistakes related to the English language were checked. We have tried, to the best of our knowledge to correct the spelling errors.
- In figure 3, left panel, the scale bar should be given.
Answer: In figure 3, left panel, the scale bar was added.
- Figures 5 and 6, the statistical method should be given.
Answer: The statistical method employed to compare the significance of the results presented in Figures 5 and 6 is now indicated. The comparison was done using the Student’s t-test.
- In figure 8, the band size label of the marker should be given
Answer: In figure 8, the closest reference band size of the marker was added.
Acknowledgment
We would like to thank the anonymous referee for his/her valuable suggestions and comments to improve the quality of this manuscript.

Reviewer 2 Report
Please summarize the introduction part into one page.
Many sentences do not have the correct punctuation and it is difficult to read the text.
English should be improved; grammar need for enhancement in many sentences and paragraphs.
All figures are needed resolution enhancement.
The figures is not in printable quality. Also, some portions of the text are losing their readability while sizing the image as per the text area. Kindly provide a better quality figure.
Please check the References in-text and end-list for uniformity in style.
Provide sufficient feedback on the main objectives of your study in the conclusion.
Author Response
- Minor spell check required.
Answer: The English language was checked and minor mistakes were corrected. Punctuation and English grammar were carefully checked.
- Please summarize the introduction part.
Answer: Considerable effort was done to shorten the introduction even though it exceeds the suggested length as we believe that the entire removal of some parts may reduce the understanding of the background information and may cause the missing of some key reference works.
- All figures need resolution enhancement. Please provide better quality figures.
Answer: The size of the figures and resolution were adjusted. We hope that the qualities of the figures are now much better as compared to those found in the previous version of the manuscript.
- Please check the references in-text and end-list for uniformity and style.
Answer: The references in-text and end-list were carefully checked. Some studies were cited more than once. That’s why the numbering in the text seems to not follow the correct order.
- Provide sufficient feedback on the main objectives of the study in the conclusion.
Answer: The conclusion part was added with a reminder of the objective of the study.
Acknowledgement
We would like to thank the anonymous referee for his/her valuable suggestions and comments to improve the quality of this manuscript.
This manuscript is a resubmission of an earlier submission. The following is a list of the peer review reports and author responses from that submission.
Round 1
Reviewer 1 Report
The present study describes an L-type lectin receptor-like kinase gene in wheat (TaLecRK-IV.1) and proposes its role in regulating plant height through the gibbirellic acid and auxin signaling pathways.
Involvement in plant height regulation was drawn by measuring transcription levels in Rht-D1b and tall (CI12633) background and by virus induced gene silencing. The identity of the Rht-D1b line used in the present study has however not been clarified. It is not clear why Yangmai 16 has been included as a control genotype vs. the dwarf line. All genotypes used in the study need to be clearly identified with genebank accession numbers. The main concern regarding the present work lays in the use of different plant materials for the different methodologies (GA treatment: Y 16, VIGS: CI12633, gene expression analyses: unidentified dwarf line and Y 16). In order to be able to draw reliable scientific conclusions the experiments (VIGS, RT-PCR, hormone treatment) need to be carried out in Rht near isogenic line series, that are genetically almost entirely identical and differ only in the different Rht alleles. Several of them are genotyped and available in public genebanks. Comparing results of gene silencing performed in one plant material with results from another plant material that has non-isogenic genetic background, is not informative.
My other comments are as follows:
The manuscript needs to be revised linguistically.
Results 2.1 Needs an introductory sentence rather than the current ‘in medias res’ opening, to make the results comprehensive. Also please specify the description in the current opening sentence, e.g., ‘single band amplified from cDNA’ is too vague and needs more detail.
Same goes for Results 2.2, please include a short introductory sentence, and define more specifically the plant material rather than just stating ‘seedlings’. Also, the current first sentence is too long and is hard to follow, needs rephrasing.
Table 1 may need to be transferred into supplementary material.
All figures need to be labelled clearly, and specifically. Such labels as ‘expression levels’, ‘control’ etc. are not suitable as they do not provide clear information about which treatment, which plant material and which gene do they refer to.
Figure 1 legend: The sentence ‘Phylogenetic tree constructed by using the deduce amino acid sequence from the genomic DNA of TaLecRK-IV.1 and other plant TaLecRK-IV.1 proteins predicted in the Genbank.’ needs to be more comprehensive. I am not sure that ‘genomic DNA of TaLecRK-IV.1’ makes sense, please rephrase.
Figure 2: Define clearly what ‘control’ means in the experiment. Which plant material, treatment, and which gene? Same goes for all the figures.
Figure 3: left panel: It is not clear what ‘knock-down’ refers to. Please replace knock-down with ‘silenced’ throughout the manuscript.
P11 L339-353 I do not see the relevance of this text in the manuscript
P12 L398 The title ’Virus-Induced Gene Silencing of TaLecRK-IV.1 ’ should be corrected as the paragraph describes the RT-qPCR methods
Reviewer 2 Report
The current manuscript revealed the role of a wheat L-type lectin receptor-like kinase gene in development. It’s novel to study the function of lectin-receptor kinase genes in plant development, especially in wheat. However, I have several major concerns about the writing and research design.
For example, the introduction didn’t provide sufficient background of the L-type lectin receptor-like kinase gene in wheat and why the authors focused on this single gene. Also, there was no primer sequence and information about how many experimental and biological replicates were used for each experiment or analysis. Moreover, which tissues do you use for the gene expression level, and what’s the expression profile of TaLecRK-IV.1, when and where did it express? Did it express in the stem? Why do you treat the Seedlings of the wheat cultivar Yangmai 16 instead of CI12633 and Rht-D1b carrying wheat line? How about the plant height of the plants treated with different phytohormones, especially the ones that up-regulated the expression of TaLecRK-IV.1?
So, I advise a rejection.